# Gut Microbiota Dysbiosis Is Associated with Elevated Bile Acids in Parkinson’s Disease

**DOI:** 10.3390/metabo11010029

**Published:** 2021-01-04

**Authors:** Peipei Li, Bryan A. Killinger, Elizabeth Ensink, Ian Beddows, Ali Yilmaz, Noah Lubben, Jared Lamp, Meghan Schilthuis, Irving E. Vega, Randy Woltjer, J. Andrew Pospisilik, Patrik Brundin, Lena Brundin, Stewart F. Graham, Viviane Labrie

**Affiliations:** 1Department of Neurodegenerative Science, Van Andel Institute, Grand Rapids, MI 49503, USA; bryan_killinger@rush.edu (B.A.K.); elizabeth.ensink@vai.org (E.E.); lubbenno@msu.edu (N.L.); mschilthuis@mcw.edu (M.S.); patrik.brundin@vai.org (P.B.); lena.brundin@vai.org (L.B.); viviane.labrie@vai.org (V.L.); 2Department of Neurological Sciences, Rush University Medical Center, Chicago, IL 60612, USA; 3Department of Epigenetics, Van Andel Institute, Grand Rapids, MI 49503, USA; ian.beddows@vai.org (I.B.); andrew.pospisilik@vai.org (J.A.P.); 4Metabolomics Department, Beaumont Health, Royal Oak, MI 48073, USA; ali.yilmaz@beaumont.org (A.Y.); Stewart.Graham@beaumont.org (S.F.G.); 5Department of Obstetrics and Gynecology, Oakland University-William Beaumont School Medicine, Rochester, MI 48309, USA; 6Integrated Mass Spectrometry Unit, Department of Translational Neuroscience, College of Human Medicine, Michigan State University, Grand Rapids, MI 49503, USA; lampjare@mail.msu.edu (J.L.); Irving.Vega@hc.msu.edu (I.E.V.); 7Department of Pathology, Oregon Health & Science University, Portland, OR 97239, USA; woltjerr@ohsu.edu; 8Division of Psychiatry and Behavioral Medicine, College of Human Medicine, Michigan State University, Grand Rapids, MI 49503, USA

**Keywords:** Parkinson’s disease, microbiome, bile acids, appendix, gut

## Abstract

The gut microbiome can impact brain health and is altered in Parkinson’s disease (PD). The vermiform appendix is a lymphoid tissue in the cecum implicated in the storage and regulation of the gut microbiota. We sought to determine whether the appendix microbiome is altered in PD and to analyze the biological consequences of the microbial alterations. We investigated the changes in the functional microbiota in the appendix of PD patients relative to controls (*n* = 12 PD, 16 C) by metatranscriptomic analysis. We found microbial dysbiosis affecting lipid metabolism, including an upregulation of bacteria responsible for secondary bile acid synthesis. We then quantitatively measure changes in bile acid abundance in PD relative to the controls in the appendix (*n* = 15 PD, 12 C) and ileum (*n* = 20 PD, 20 C). Bile acid analysis in the PD appendix reveals an increase in hydrophobic and secondary bile acids, deoxycholic acid (DCA) and lithocholic acid (LCA). Further proteomic and transcriptomic analysis in the appendix and ileum corroborated these findings, highlighting changes in the PD gut that are consistent with a disruption in bile acid control, including alterations in mediators of cholesterol homeostasis and lipid metabolism. Microbially derived toxic bile acids are heightened in PD, which suggests biliary abnormalities may play a role in PD pathogenesis.

## 1. Introduction

Parkinson’s disease (PD) is the second most common neurodegenerative disease, affecting about 1% of the population over 60 years of age [1]. PD is clinically characterized by motor and non-motor symptoms. Some non-motor features of PD begin many years before the onset of motor symptoms; one of the first symptoms is constipation, pointing toward an early involvement of the gastrointestinal (GI) tract [2]. Aggregated α-synuclein (α-syn), a pathological hallmark of PD, is apparent in the GI tract of prodromal PD patients [3,4]. α-syn aggregates in the gut of experimental animal models and has been reported to propagate to the brain and induce nigral neurodegeneration and PD-like motor and non-motor dysfunctions [5,6]. Recently, the appendix has been implicated as one GI tract location that could contribute to PD pathogenesis [7]. The appendix contains an abundance of aggregated α-syn, particularly in enteric nerves, with PD patients having higher amounts of these aggregates [7]. The removal of the appendix was associated with a decreased risk for PD in some, but not all, epidemiological studies [7,8,9]. This suggests that the appendix may be an important tissue to study to advance our understanding of some of the earliest events in this disease.

The appendix is an immunological organ that also acts as a storehouse for the gut microbiota [10,11]. The gut microbiota and their metabolites are increasingly being recognized as crucial for brain health [12]. Numerous studies report microbiota changes in the stool of PD patients as compared to healthy controls [13,14,15,16]. The appendix contains rich microbial flora, which differ from that of the rectum and stool [17,18]. Importantly, the appendix has an anatomically shielded microbiome and can modulate and repopulate the microbiome in the rest of the large intestine [19,20]. Consequently, changes in the appendix microbiome may have a widespread effect on the microbiota of the intestine, which may be reflected in changes in stool microbiota. Furthermore, inflammation in the periphery and the brain has been proposed to have a central role in PD, and the microbiome and the host immune system have a bidirectional effectual relationship [12,21,22]. Microbiota can modify inflammatory responses and immunity in the gut, and the lymphoid tissue of the appendix is particularly relevant since it is especially rich in lymphocytes compared to the rest of the GI tract [11]; microbial metabolites have direct access to the immune cells within the lymphoid follicles of the appendix [23,24,25]. Thus, the dysregulation of the appendix microbiota may be involved in PD, but this has yet to be examined.

One important function of the gut microbiota is their involvement in the biotransformation of bile acids. Bile acids aid in the absorption of dietary lipids and affect glucose homeostasis, inflammation, gastrointestinal functions, as well as blood–brain barrier integrity and signaling in the brain [26,27,28]. In the liver, primary bile acids are synthesized from cholesterol. After being released from the gall bladder, most primary bile acids are reabsorbed in the ileum for transport back to liver [26]. The remaining primary bile acids that enter the large intestine are converted by the microbiota (largely by those in *Clostridium* clusters *XIVa* and *XI*) into secondary bile acids—deoxycholic acid (DCA), lithocholic acid (LCA), and ursodeoxycholic acid (UDCA) [26]. Lithocholic acid and DCA are hydrophobic bile acids that are cytotoxic at elevated physiological concentrations [29,30]. Increases in LCA and DCA have been implicated in intestinal inflammation, liver injury, cholestasis, and gallstone formation [29,31,32,33]. Whether there are hydrophobic bile acid changes that impact PD risk is unknown.

To test our hypothesis that the appendix will show microbiota changes in PD, with an impact on the associated transcription and protein homeostasis in the gut, we performed a multi-omic analysis of the appendix microbiome in PD patients and controls. We performed a multi-omic analysis of the appendix microbiome, including metabolomics, proteomics, and transcriptomics in PD patients and controls.

## 2. Results

### 2.1. Microbiota Changes in PD Appendix

Here, we performed a comprehensive microbiome analysis in the appendix tissue of PD and controls (*n* = 12 and 16 respectively; Appendix A) using metatranscriptomic sequencing, which profiles the functionally active microbiota. We had on average 14,288,947 ± 5,008,507 reads per sample, and found transcripts for 65 genera, 37 families, 20 orders, 15 classes, and 9 phyla. We did not find changes in the richness of microbiota communities between PD and controls at any taxonomic level (alpha diversity; Appendix A). The appendix of PD and controls also had a similar overall microbial community composition (beta-diversity; Appendix A). The most abundant bacteria in the appendix overall were *Lachnospiraceae*, *Ruminococcaceae*, *Porphyromonadaceae*, *Enterobacteriaceae*, and *Bacteroidaceae*, together accounting for 68.6% of the relative family abundance (Appendix A). We also compared the appendix microbial community identified to that found in surgically isolated, healthy appendix tissues from a previous study and found significant correlation (order level: R = 0.91, *p* < 10^−14^; family level: R = 0.26, *p* < 0.05; Pearson’s correlation).

In an analysis examining the abundance of microbial taxa, differences between the appendix microbiota of PD and controls were observed at all taxonomic levels (*q* < 0.05; Figure 1a; Appendix A). The most significant change in the PD appendix microbiota relative to controls was in the order of *Clostridiales*, particularly an increase in *Peptostreptococcaceae* and *Lachnospiraceae*. The appendix of PD patients also had a prominent increase in *Burkholderiales* and decrease in *Methanobacteriales*. Furthermore, the PD appendix had decreases in genera *Odoribacter*, *Clostridium*, *unclassified Sutterellaceae*, and *Escherichia*.

### 2.2. Proteomic Changes in the PD Gut

Then, we examined the major microbial metabolic pathways altered in the PD appendix. We report that the most significant microbial pathway change was a loss of fatty acid metabolism and the dysregulation of the lipid metabolism (*q* < 0.1; Figure 1b; Appendix A). Considering the prominent disruption of the lipid metabolic pathways in the microbiome of PD patients, we analyzed the human proteome in the appendix tissue of PD and controls. Pathway analysis of the proteomic data highlighted a reduction in proteins affecting lipid metabolism (*q* < 0.05; Figure 2a; Appendix A). There was also a dysregulation of pathways involved in protein localization, antigen presentation, glycolysis, and immune activity in the PD appendix (Figure 2a). Overall, the proteomic changes affecting lipid homeostasis in the PD appendix corroborate the microbial pathway alterations affecting lipids in patients.

In addition, we profiled the proteome of the ileum of PD patients and compared it to that of controls with the aim of identifying significantly altered metabolic pathways. We found that the PD ileum had a decrease in lipid metabolism, as observed in the PD appendix (*q* < 0.05; Figure 2b). In the PD ileum, there was also a dysregulation of antigen processing and presentation, immune activation, glycolysis, and actin filament organization (*q* < 0.05; Figure 2b). We then determined the proteins most consistently altered in the PD ileum and appendix [34]. We found that both the PD ileum and appendix had a strong decrease in fatty acid binding protein 6 (FABP6), the intracellular bile acid transporter involved in returning bile acids to enterohepatic circulation (Figure 2c).

### 2.3. Microbiota-Driven Bile Acid Changes in the Gut of PD Patients

In addition to identifying perturbations in the lipid metabolism in the PD appendix, we also report changes in microbiota responsible for generating hydrophobic secondary bile acids (*Clostridium* cluster *XI* and *Burkholderiales*). Consequently, we measured the bile acid levels in the PD appendix. We accurately quantified 15 bile acids in PD and healthy controls (*n* = 15 and 12, respectively) (Figure 3a; Appendix A) and found an 18.7-fold increase in LCA and a 5.6-fold increase in DCA in the PD appendix relative to the controls (*p* < 0.05; Figure 3; Appendix A). We did not observe any changes in primary bile acids or total bile acid levels in the PD appendix, but elevated concentrations of secondary bile acids produced by the microbiota (*p* < 0.05). In addition, we measured the concentrations of bile acids in the ileum of PD patients (*n* = 20 PD and 20 controls) and once again report no changes in the primary bile acid concentrations but do show a significantly marked increase in LCA in the PD ileum as compared to the controls (3.6-fold; Figure 3; Appendix A).

### 2.4. Bile-Associated Transcriptomic Changes in the PD Gut

Since we found an abundance of microbiota-derived bile acids in PD, we investigated whether PD patients had a differential expression of genes important for bile acid biosynthesis, signaling, and transport. In the ileum of PD and controls, we examined the transcript levels of nuclear receptors that regulate bile acid and cholesterol homeostasis (*FXR* and *LXR*), bile acid-sensing receptor (*TGR5*), transporters for bile reabsorption (*ASBT*, *OSTα/OSTβ*, *FABP6*), and transporters for cholesterol reabsorption and efflux (*NPC1L1* and *ABCG5/ABCG8*, respectively). We also examined the liver tissue of PD and controls for transcript levels of these genes or their liver-specific functional equivalents (*NTCP* and *FABP1*), as well as enzymes for bile acid biosynthesis in the liver (*CYP7A1* and *CYP27A1*). We found that the PD ileum had significantly elevated levels of gene transcripts involved in cholesterol homeostasis and transport (*p* < 0.05; Figure 4a; Appendix A). In the liver, we did not observe changes in bile acid-related transcripts (Figure 4b; Appendix A). In conclusion, the human transcriptomic and proteomic changes in the PD gut are consistent with a disruption in bile acid control, including alterations in mediators of cholesterol homeostasis and lipid metabolism.

## 3. Discussion

Our metatranscriptomic analysis revealed significant differences in the appendix microbiota of PD patients, which are closely related to bile acid dysregulation in the gut. The microbial taxa and pathway alterations we identified led us to further hypothesize that bile acid metabolism might be directly associated with the etiopathogenesis of PD. Thus, we investigated the potential changes in bile acid metabolism in PD using metabolomics, proteomics, and transcriptomics. We report significant increases in the concentrations of secondary bile acids in the human appendix and ileum (*p* < 0.05). Furthermore, proteomic and transcriptomic analysis support the dysregulation of lipid metabolism and cholesterol homeostasis in the gut of PD patients. 

We found significant microbiota differences in the PD appendix using a metatranscriptomic approach. Unlike the prior 16S rRNA and metagenomic studies of the PD microbiome, our metatranscriptomic study has the advantage of capturing only active species. The appendix microbial community identified was similar to that found in surgically isolated, healthy appendix tissues in a previous study, demonstrating reliability in our methods [17]. Another strength of our study is the unique location of the survey; microbiota shifts in the appendix are particularly relevant given the role of the appendix in affecting the microbiota in other intestinal regions [11]. Differences between the appendix and fecal microbiomes limit comparison with previous studies of the PD microbiota; however, an increase in *Clostridiales* cluster *XI* has been observed in PD stool previously [35]. Nonetheless, one limitation of our study is that it does not accurately distinguish microbiota changes which may contribute to the causes of PD from those that are consequences of the disease. Both constipation (a common symptom of PD) and PD medication may influence the gut microbiota [35]. Though we cannot exclude constipation as a contributing factor, it is unlikely to explain all the changes observed, since increases in both LCA and DCA increase colonic peristalsis which decreases the fecal transit time [36,37,38]. Although we controlled for age and sex, additional clinical variables such as medication or the duration of disease were not available for this dataset but would be valuable to test for correlations; we aim to consider this in future work.

The microbiome clearly plays a role in the health of the nervous system [12,39,40], and our results implicate bile acid metabolism as another pathway by which changes in the microbiota may contribute to the etiology and pathogenesis of PD. Bacterial species that are responsible for the production of secondary bile acids in the large intestine were elevated in the PD appendix. We report significant increases in *Burkholderiales* which is a bacterial genus with a broad environmental distribution [41]. *Burkholderia* can cause severe inflammation in immunocompromised individuals [41] and produce kynurenine and quinolinate [42], which are proinflammatory metabolites associated with symptom severity in PD [43]. *Burkholderia* has also been reported to infect the brain [44,45]. Of particular interest, *Burkholderia* species encode the rate-limiting enzyme for secondary bile acid synthesis (bile-acid dehydratase) [46], highlighting the fact that the appendix microbiome of PD sufferers has an enrichment of microbiota that metabolize bile acids.

In line with the microbiota shifts as observed in the PD appendix, we found an increase in the secondary bile acids LCA and DCA in the ileum of PD patients. LCA and DCA are highly hydrophobic, and their increase can have pro-inflammatory and direct cytotoxic effects [27,29,30]. These effects could propel the accumulation of pathological α-syn aggregates, which could potentially propagate from the gut to the brain through retrograde transport [47,48]. Changes in the appendix are especially relevant since the appendix holds an abundance of α-syn even in healthy individuals [11]. Alterations in the microbiome have been correlated with changes in melatonin synthesis along with serum DCA levels [49]; melatonin, which is highly produced in the appendix, can suppress the accumulation of toxic α-syn and may contribute to PD etiopathogenesis. Thus, local melatonin levels in the PD appendix may impact both the levels of secondary bile acids and α-syn aggregation [50]. In addition, since bile acids can modulate the immune response [32,33,51,52], LCA and DCA may contribute to T cell activation, which has been implicated in PD pathogenesis [53], and/or the microbial-driven bile acid changes may be a response to α-syn aggregation and concurrent inflammation in the gut of PD sufferers. Further in vivo models are essential to elucidate the effects of secondary bile acid metabolism on PD pathology.

Our study provides some insight into the potential causes behind the changes in the microbiota and bile acid composition in PD, which merit further investigation. In the PD patients, there was no evidence of an overproduction of primary bile acids: primary bile acids and the total bile acid pool size in the PD ileum remained similar to the controls, and we did not detect transcript abnormalities in enzymes responsible for primary bile acid synthesis in the PD liver. However, PD patients may have impaired bile acid reuptake in the ileum, as indicated by the prominent decrease in FABP6, a protein responsible for the efficient transport of primary bile acids through enterocytes for recirculation [54]. Thus, disrupted bile acid transport in the ileum and increased bile-metabolizing bacteria in the large bowel combined could be responsible for the elevated levels of secondary bile acids as observed in PD. These disruptions may also contribute to the observed abnormalities in cholesterol homeostasis in the PD gut, as demonstrated by the transcriptional increase in cholesterol transporters and the proteomic disruption of lipid metabolism pathways. It is also worth noting that changes in bile acids can in turn modulate the composition of the microbiota. Though this study does not delineate which changes occur first in PD, the bi-directional relationship between the microbiota and bile acid composition creates a system which, once triggered, may lead to a self-reinforced condition of dysbiosis, peripheral inflammation, and α-syn aggregation. 

Targeting the appendix microbiome and bile acids may be an innovative approach for future therapeutics. Our results support microbiome transplantation as a potential treatment for PD [55]. In addition, preventing the damaging effects of LCA and DCA with the hydrophilic, anti-inflammatory bile acid UDCA could benefit PD patients. UDCA has been shown to counteract the effects of hydrophobic secondary bile acids in the liver and gallbladder [56,57]. UDCA also has neuroprotective effects [58] and is currently being tested in clinical trials for PD [59]. 

In sum, our findings provide a novel look into the appendix microbiota in PD and demonstrate microbially mediated bile acid disturbances (Appendix A). Though further investigation is needed, bile acids could play a key role at the intersection of microbiome dysbiosis, inflammation, and α-syn misfolding. Considering the relative accessibility of GI the tract and existing therapies for bile acid-related disorders, targeting microbial-derived secondary bile acids may be a new avenue for the earlier diagnosis and alleviation of PD symptoms.

## 4. Materials and Methods 

### 4.1. Human Tissue Samples

The human appendix, ileum, and liver tissue from PD patients and controls was obtained from the Oregon Brain Bank with information on demographics (age, sex), tissue quality (postmortem interval), and pathological staging for each individual (Appendix A). Tissue was collected and flash frozen within an average postmortem interval of 14 h and shipped frozen on dry ice. PD cases were selected based on pathologically confirmed presence of brain Lewy body pathology and the loss of midbrain neurons, and control individuals were selected for the absence of such pathology. Cases and controls were balanced by sex and age across groups. All human postmortem tissue work had approval from the Van Andel Institute ethics committee (IRB #15025).

### 4.2. Metatranscriptomic Analysis of PD Appendix Microbiota

To profile the functional microbiome changes in the PD appendix, we performed a metatranscriptomic analysis of the appendix of 12 PD and 16 controls. Frozen appendix tissue (~20 mg) was homogenized using a Covaris cryoPREP pulverizer and then in 1 mL of TRIzol (Life Technologies, Carlsbad, CA, USA) with a ceramic bead-based homogenizer (Precellys, Bertin Instruments, Montigny-le-Bretonneux, France). Total RNA was isolated according to the TRIzol manufacturer’s instructions, treated with RNase-free DNase I (Qiagen, Hilden, Germany) at room temperature for 30 min, and cleaned up with the RNeasy Mini Kit (Qiagen). Total RNA yield and quality was determined using a NanoDrop ND-1000 (Thermo Fisher Scientific, Waltham, MA, USA) and an Agilent Bioanalyzer 2100 system (Agilent Technologies, Santa Clara, CA, USA). Libraries were prepared by the Van Andel Genomics Core from 300 ng of total RNA using the KAPA RNA HyperPrep Kit with RiboseErase (v1.16; Kapa Biosystems, Wilmington, MA, USA). RNA was sheared to 300–400 bp. Prior to PCR amplification, cDNA fragments were ligated to NEXTflex dual adapters (Bioo Scientific, Austin, TX, USA). The quality and quantity of the finished libraries were assessed using a combination of Agilent DNA High Sensitivity chip (Agilent Technologies, Inc.), QuantiFluor dsDNA System (Promega Corp., Madison, WI, USA), and Kapa Illumina Library Quantification qPCR assays (Kapa Biosystems, Wilmington, MA, USA). Individually indexed libraries were pooled, and 100 bp, single-end sequencing was performed on an Illumina NovaSeq6000 sequencer using an S1 100 cycle kit (Illumina Inc., San Diego, CA, USA), with all libraries run on a single lane to return an average depth of 37 million reads per library. Base calling was done by Illumina RTA3, and the output of NCS was demultiplexed and converted to a FastQ format with Illumina Bcl2fastq v1.9.0.

The preprocessing of metatranscriptomic data involved the removal of sequencing adapters and low-quality bases from sequencing reads using Trim Galore (v0.5.0). The transcriptomic data were aligned to human genome (GRCh38/hg38) with the twopassMode basic algorithm in STAR (v2.5.2b) [60]. Reads that did not align to the human genome (STAR option outReadsUnmapped) were then input to MetaPhlAn2 (v2.7.7) [61], which gives kingdom to species-level resolution (db_v20). We then performed the functional profiling of the microbial community for the same non-human reads using HUMAnN2 (v0.11.1) [62] with the UniRef90 database. Pathway abundance data were normalized to counts per million using the inbuilt HUMAnN2 functionality. To test for differential taxa abundance between the PD and control, proportional microbial compositional data from MetaPhlAn2 were imported into R (v3.6) and converted back to counts for all taxa-level ids (features). To perform statistical analysis, we used the cumulative sum scaling normalization and the zero-inflated Gaussian mixture model from the bioconductor package metagenomeSeq (v1.28.0) [63]. Feature and pathway abundance data were examined using the fitZig function to determine the microorganisms and pathways related to PD, adjusting for age, sex, postmortem interval, and RIN. *p*-values were derived from the empirical Bayes moderated F-statistic and adjusted for multiple testing correction using the Benjamini–Hochberg method, with *q* < 0.05 set as the threshold for statistical significance.

### 4.3. Mass Spectrometry and Proteomics Analysis

Here, quantitative proteomic analysis was performed to determine the host biological pathways altered in the PD appendix and ileum. For this analysis, we used existing proteomic data from the PD and control appendix (*n* = 3 individuals/group; PXD015079) and generated new proteomic data for the PD and control ileum (*n* = 4 individuals/group). Mass spectrometry for the appendix and ileum samples was performed using the same protocol by the Integrated Mass Spectrometry Unit at Michigan State University. The wet tissue weight of each sample (~30 mg tissue/sample) was measured and a 5-fold lysate buffer (20 mM Tris Base (pH 7.4), 150 mM NaCl, 1 mM EGTA, 1 mM EDTA, 5 mM sodium pyrophosphate, 30 mM NaF, 1X Halt Protease Inhibitor Cocktail (Thermo Fisher Scientific) was used to homogenize the tissue on ice with a tissue grinder (Tissue Master 125, Omni International). The homogenate was centrifuged at 18,407× *g* for 10 min at 4 °C and the supernatant was retained. Protein concentration in each sample was determined using a BCA assay (Pierce BCA Protein Assay, Thermo Fisher Scientific). Protein lysates (10 μg) were denatured using 25 mM ammonium bicarbonate/80% acetonitrile and incubated at 37 °C for 3 h. The samples were dried and reconstituted in 50 μL of 25 mM ammonium bicarbonate/50% acetonitrile/trypsin/LysC solution (1:10 and 1:20 *w/w* trypsin:protein and LysC:protein, respectively) and digested overnight at 37 °C. The samples were dried and reconstituted in 50 μL of 25 mM ammonium bicarbonate/5% acetonitrile. 

Samples were loaded onto an UltiMate 3000 UHPLC system with online desalting. Each sample (10 μL) was separated using a C18 EASY-Spray column (2 μm particles, 25 cm × 75 μm ID) and eluted using a 2 h acetonitrile gradient into a Q-Exactive HF-X mass spectrometer. Data-dependent acquisition for the full MS was set using the parameters described in the supplementary methods. Each sample was run in triplicate. The mass spectra from each technical replicate were searched against the Uniprot human database (filtered proteome_3AUP000005650) using the Label-free quantification (LFQ) method in Proteome Discoverer (v. 2.2.0.388, 2017). Data-dependent acquisition for the full MS was set using the following parameters: resolution 60,000 (at 200 *m*/*z*), Automatic gain control (AGC) target 3 × 10^6^, maximum Injection time (IT) 45 s, scan range 300 to 1500 *m*/*z*, dynamic exclusion 30 s. Fragment ion analysis was set with the following parameters: resolution 30,000 (at 200 *m*/*z*), AGC target 1 × 10^5^, maximum IT 100 ms, TopN 20, isolation window 1.3 *m*/*z*, Normalized collision energy (NCE) at 28. Each sample was run in triplicate. The mass spectra from each technical replicate were searched against the Uniprot human database (filtered-proteome_3AUP000005650) using the LFQ method in Proteome Discoverer (v. 2.2.0.388, 2017) set as follows: at least 2 peptides (minimum length = 6, minimum precursor mass = 350 Da, maximum precursor mass 5000 Da), tolerance set to 10 ppm for precursor ions and 0.02 Da for fragment ions (b and y ions only), dynamic modification was set for methionine oxidation (+15.995 Da) and N-terminus acetylation (+42.011 Da), target FDR (strict minimum value 0.01), Delta Cn minimum value 0.05). LFQ was calculated using the following parameters: the ratio calculation was set to pairwise ratio based, maximum allowed fold change 100, ANOVA (background based).

The technical replicates from each biological sample were pooled to perform diagnosis comparisons, using a non-nested test. Proteins were quantified using the pairwise peptide ratio information from extracted peptide ion intensities. Only proteins with abundances recorded in at least 50% of samples were considered. Proteins with a log fold change between groups exceeding ±0.2 were considered altered. The pathway analysis of proteins altered in the PD appendix and ileum was performed using g:Profiler [64], with networks determined by EnrichmentMap and clustered by AutoAnnotate in Cytoscape (v3.7.1) [65]. To identify the proteins that were most altered, we first determined the proteins that exhibited significant changes and had the same direction of change in both the PD appendix and ileum. These altered proteins were ranked by log fold change, ranking separately for appendix and ileum. We then determined the proteins most consistently altered using the aggregateRanks function from the RobustRankAggreg package (v1.1) [34]. 

### 4.4. Bile Acid Sample Preparation and Metabolite Quantification

Liquid chromatography–mass spectrometry (LC–MS) was used to measure the primary and secondary bile acids in the appendix and ileum of PD patients and controls (appendix, *n* = 12 controls, 15 PD; ileum, *n* = 20 controls, 20 PD). Tissues (25 mg) were homogenized using a bead homogenizer at 5500 rpm for 30 s in the 300 µL of extraction solvent (85% ethanol and 15% phosphate-buffered saline solution). Samples were then sonicated at 4 °C for 10 min. Proteins and other impurities were removed by centrifugation at 13,000× *g* for 30 min at 4 °C. The supernatant was collected and 10 µL was loaded onto the Biocrates Bile Acid kit (Biocrates Life Sciences). Data were acquired using an Acquity I-class UPLC (Waters) coupled with a Xevo TQ-S mass spectrometer (Waters). All specimens were acquired in accordance with the protocol for the Biocrates Bile Acids kit. Bile acid concentrations (nmol per gram of tissue weight) were calculated utilizing the Biocrates MetIDQ software and TargetLynx (Waters). For the group-level analyses, the concentration of the bile acid and its glycine and taurine conjugates were summed. For example, the LCA group was the sum of LCA, GLCA, and TLCA, and the DCA group was the sum of DCA, GDCA and TDCA. For the all-primary, all-secondary, and total bile acid analyses, the respective bile acids and their conjugates were summed.

Bile acid data were normalized by log10 transformation, as previously described [66]. Bile acid changes in the PD appendix and ileum were determined by multivariate robust linear regression models with empirical Bayes from the limma (v3.30.13) statistical package [67], adjusting for age, sex, and postmortem interval.

### 4.5. qPCR Analysis of Gene Transcripts Involved in Bile Acid and Cholesterol Homeostasis

We examined the transcriptional changes of genes involved in bile acid transport and cholesterol homeostasis in the PD ileum (*n* = 6 controls, 8 PD) and liver (*n* = 6 controls, 6 PD). Samples from the PD and healthy controls were matched for sex, age, and postmortem interval. Tissue (30–50 mg per sample) was homogenized in 1 mL TRIzol with a handheld homogenizer (Biospec) for the rileum and with Precellys bead tubes (Bertin Corp, Rockville, MD, USA) for the liver. Following standard TRIzol RNA extraction, samples were treated with DNase (Qiagen) for 30 min. RNA cleanup was performed using a RNeasy column (Qiagen) according to the manufacturer’s instructions, with the addition of two 75% ethanol washes. Isolated RNA quantity was determined with a NanoDrop 2000 spectrophotometer, and RNA integrity was confirmed with an Agilent 2100 Bioanalyzer. RNA was converted to cDNA using a High Capacity cDNA kit (Applied Biosystems, Carlsbad, CA, USA). Samples were analyzed by qPCR with TaqMan reagents (Applied Biosystems; Appendix A), using 25 ng of cDNA per qPCR reaction. Samples were run in triplicate and the results were normalized to plate standardization controls. The delta delta CT values of gene transcripts were used to determine the statistical changes in the ileum and liver of PD relative to the controls, normalized to housekeeping control genes (*β-actin* and HPRT for the liver, β-actin or villin for ileum). Statistical analysis was performed using one-way ANOVA with *p* <0.05 considered to be significant changes.

## Figures and Tables

**Figure 1 metabolites-11-00029-f001:**
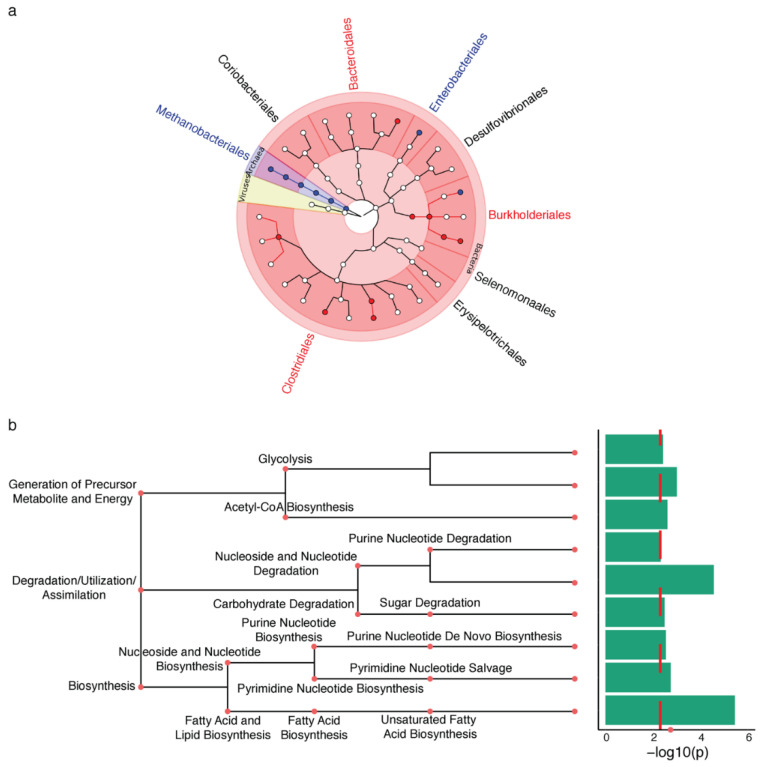
The Parkinson’s disease (PD) appendix exhibits shifts in the active microbiota that affect lipid metabolism. Metatranscriptomic analysis was used to determine the changes in the functional microbiota in the PD appendix (*n* = 12 PD, 16 controls). (**a**) Microbiota changes in the PD appendix. Metatranscriptome data were analyzed by MetaPhlAn2 and a zero-inflated Gaussian mixture model in metagenomeSeq, adjusting for age, sex, RNA integrity number (RIN), and post-mortem interval. Results are displayed using GraPhlAn, showing the taxonomic tree with the kingdom in the center, and branching outwards to phylum, class, order, family, and genus. Microbial taxa highlighted in red are increased in PD, and blue are decreased in PD (*q* < 0.05, metagenomeSeq). (**b**) Microbiota metabolic processes are altered in the PD appendix. Top microbial pathways are altered in PD as identified by HumanN2. Red dashed line denotes *q* < 0.1 pathways as determined by metagenomeSeq.

**Figure 2 metabolites-11-00029-f002:**
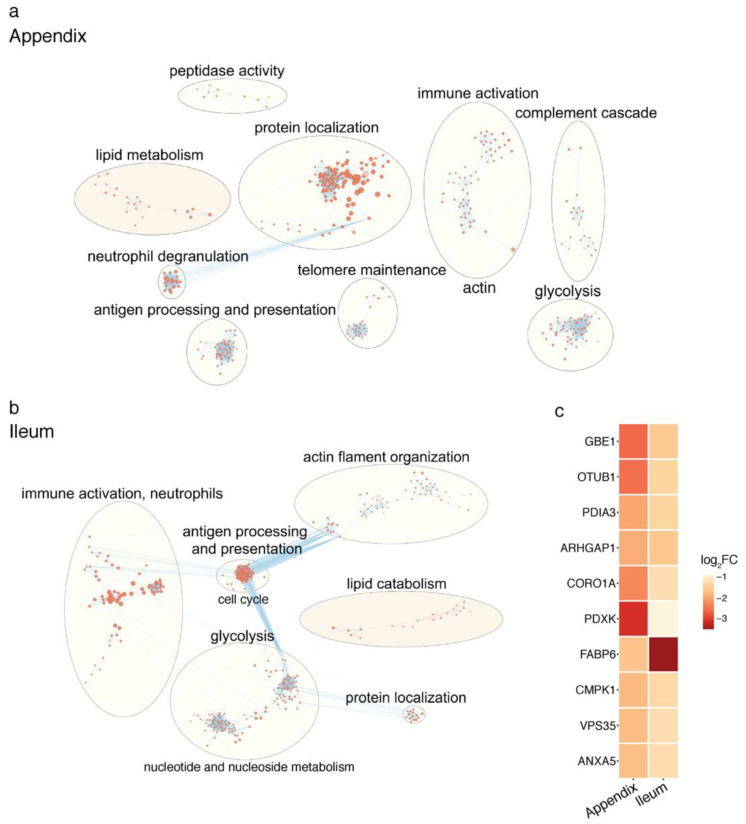
Proteomic analysis identifies the altered lipid metabolism pathways in the PD appendix and ileum. Pathway enrichment analysis of proteomic changes in the PD appendix relative to controls (*n* = 3 PD, 3 controls) (**a**) and in PD ileum relative to controls (*n* = 4 PD, 4 controls) (**b**). Pathway analysis of quantitative proteomic data was performed using g:Profiler. Nodes are pathways altered in the PD appendix that were clustered into functionally similar networks by EnrichmentMap (nodes are *q* < 0.05 pathways, hypergeometric test). Node size represents the number of genes in the pathway gene set, and the edges connect pathways with similar gene sets (0.7 similarity cutoff). The lipid metabolism pathway network is highlighted in peach. (**c**) Top 10 proteins that were most consistently altered in the PD appendix and ileum. Heatmap showing the proteins ranked as the most consistently disrupted in the PD appendix and ileum, as determined by a robust ranking algorithm. Log fold change is shown, and red signifies greater disruption in PD.

**Figure 3 metabolites-11-00029-f003:**
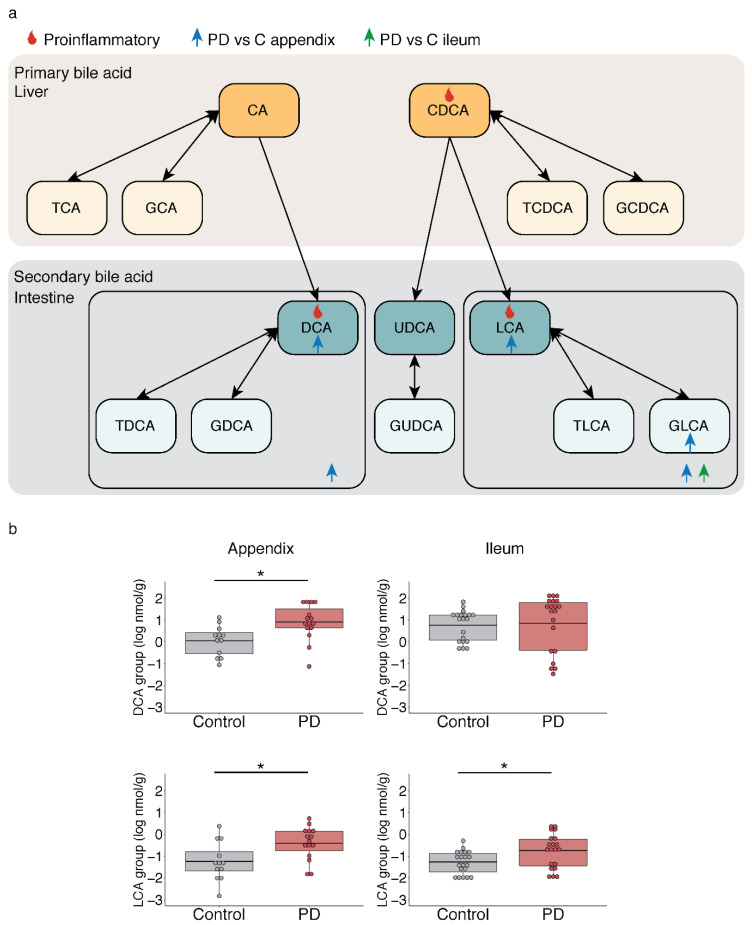
Increase in the microbiota-derived secondary bile acids in the appendix of PD patients. Bile acid analysis was performed by liquid chromatography–mass spectrometry in the PD and control appendix (*n* = 15 PD, 12 controls) and ileum (*n* = 20 PD, 20 controls). Bile acid changes were determined by robust linear regression, controlled for age, sex, and postmortem interval. (**a**) Illustration of the bile acid changes identified in this study and the bile acid pathway. Primary bile acids are generated in the liver and secondary bile acids are produced by microbiota in the intestine. In the secondary bile acid section of the image, boxes highlight the DCA and LCA groups (DCA, LCA and their respective conjugates). Bile acids increased in the PD appendix or PD ileum, relative to controls, are marked by a blue and green arrow, respectively. The flame symbol denotes hydrophobic bile acids that have proinflammatory effects when elevated. (**b**) Secondary bile acid changes in the PD appendix and PD ileum. The boxplot center line represents the mean, the lower and upper limits are the first and third quartiles (25th and 75th percentiles), and the whiskers are 1.5× the interquartile range. * *p* < 0.05, robust linear regression.

**Figure 4 metabolites-11-00029-f004:**
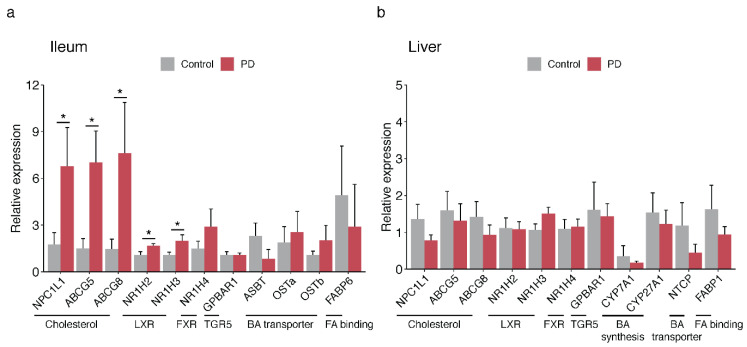
Dysfunctional cholesterol and lipid metabolism in the PD ileum. Transcript levels of genes in the ileum (**a**) and liver (**b**) affecting the abundance of cholesterol and bile in the enterohepatic circulation. Transcript levels of genes affecting cholesterol and bile acid homeostasis (*NR1H2*, *NR1H3*, *NR1H4*, *GPBAR1*) and their transport and reabsorption into the enterohepatic circulation (*NPC1L1*, *ABCG5*, *ABCG8*, *ASBT*, *OSTα*, *OSTβ*, *FABP6*) were examined in the ileum. In the liver, the transcript levels of these genes or the equivalent bile acid transporters (*NTCP*, *FABP1*) were examined, as well as the rate-limiting enzymes for bile acid production (*CYP7A1*, *CYP27A1*). Transcript levels were analyzed by qPCR and normalized to housekeeping genes (*villin1*, *β-actin*). The relative expression ± s.e.m in the ileum (*n* = 8 PD, 6 controls) and liver (*n* = 6 PD, 6 controls). * *p* < 0.05, one-way ANOVA.

## Data Availability

All sequencing data used in this study are available from the NCBI Gene Expression Omnibus (GEO) database under the accession number GSE135743. The mass spectrometry proteomics data of human ileum has been deposited to the ProteomeXchange Consortium via the PRIDE partner repository with the dataset identifier PXD020988. The human appendix proteomic data was obtained from PXD015079.

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
