# Peer review of "Gut Microbiota Dysbiosis Is Associated with Elevated Bile Acids in Parkinson’s Disease"

_metabolites, 2021, doi:10.3390/metabo11010029_

Round 1

Reviewer 1 Report

The work presented is very interesting. My comments are as follows:

Abstract

- Some information regarding the study design is missing and should be added.

Introduction

- Line 37: delete “The introduction”

- Line 37: in order to know how common PD is, some epidemiological data are needed.

- Lines 40 and 41: please, use either “α-syn” or  “α-Syn”.

- Line 42: please, add which types of experimental models (in vitro? animal models?)

- Which is the relationship between “α-syn” and appendix?

- Line 47 and further: as far as I know, “microbiome” is referred to the genetic burden that microbiota provides to the host. However, it is commonly used to address “microbiota”, despite of being wrongly used. The right word to address the microorganisms (not their genetic burden) is “microbiota”. Please revise the whole manuscript.

- Line 51: how changes in the stool of PD patients can be related to microbiota in appendix?

- Line 54: gut microbiota (not exclusively gut microbiota in appendix) have an intimate cross-talk with the immune system in the gut.

- The hypothesis is not clearly defined. Neither the objective. Authors should not report results or conclusions at the end of the introduction; only hypothesis and objective.

Results

- Line 87: please, avoid citations in this section. Any explanation of the results should be included in the discussion section.

Discussion

- Limitations and strengths of the study are needed.

Methods

- A first paragraph describing the study design is needed. Details about selection criteria is also needed. Also, preservation methods of tissues in the extraction place and transport should be stated.

- Was the sample size calculated beforehand? If now, which was the rationale for using that and no other sample size?

Author Response

Review 1

The work presented is very interesting. My comments are as follows:

Abstract

- Some information regarding the study design is missing and should be added.

Response: We revised the abstract to add details on the study design including the sample size for the major experiments. We also added that ileum was considered in the bile acid quantification.

The following changes were made to the manuscript:

Page 1 Line 23-25: We investigate changes in the functional microbiome in the appendix of PD patients relative to controls (n=12 PD, 16 C) by metatranscriptomic analysis.

Page 1 Line 26-28: We then quantitatively measure changes in bile acid abundance in PD appendix relative to controls (n=15 PD, 12 C) and the ileum (n=20 PD, 20 C).

Page 1 Line 29-30: Further proteomic and transcriptomic analysis in the appendix and ileum corroborated these findings

Introduction

- Line 37: delete “The introduction”

Response: Deleted as requested

- Line 37: in order to know how common PD is, some epidemiological data are needed.

Response: We have added the following epidemiological data to the introduction:

Page 1 Lines 37-38 : Parkinson’s disease (PD) is the second most common neurodegenerative disease, affecting about 1% of the population over 60 years of age.

- Lines 40 and 41: please, use either “α-syn” or “α-Syn”.

Response: We thank the reviewer for this attention to detail. We note that ““α-syn” is used consistently throughout the manuscript but was capitalized here to follow capitalization norms for the start of the sentence. We have revised “α-Syn” to “α-syn” and leave this to the discretion of the editor (page 1, line 42)

- Line 42: please, add which types of experimental models (in vitro? animal models?)

Response: We have revised this sentence to be more specific by including “animal models”:

Page 1 Line 42-44: “α-Syn that aggregate in the gut of experimental animal models have been reported to propagate to the brain and induce nigral neurodegeneration and PD-like motor and non-motor dysfunctions.”

- Which is the relationship between “α-syn” and appendix?

Response: We thank the reviewer for the opportunity to clarify the connection between α-syn and the appendix. We have added the following text to the manual to describe how the PD appendix contains an abundance of α-syn, which may be associated with disease risk:

Page 1-2 Line 45-48: “The appendix contains an abundance of aggregated α-syn, particularly in enteric nerves, with PD patients having higher amounts of these aggregates [7]. Removal of the appendix was associated with a decreased risk for PD in some, but not all, epidemiological studies [7-9].”

- Line 47 and further: as far as I know, “microbiome” is referred to the genetic burden that microbiota provides to the host. However, it is commonly used to address “microbiota”, despite of being wrongly used. The right word to address the microorganisms (not their genetic burden) is “microbiota”. Please revise the whole manuscript.

Response: We thank the reviewer for helping us make our language more precise. We have revised “microbiome” to “microbiota” throughout the manuscript in places where we specifically refer to the microorganisms themselves. We kept “microbiome” in places where we refer more holistically to the micro-organisms, their genetic burden, and the environment, as suggested by [1].

  1. Marchesi, J.R.; Ravel, J. The vocabulary of microbiome research: a proposal. Microbiome 2015, 3, 31, doi:10.1186/s40168-015-0094-5.

- Line 51: how changes in the stool of PD patients can be related to microbiota in appendix?

Response: We have revised this paragraph to clarify how the microbiota in the appendix can modulate the microbiota of the rest of the intestine. The following text has been added to the manuscript:

Page 2 Line 54-58: “The appendix contains rich microbial flora, which differ from that of the rectum and stool [12,13]. Importantly, the appendix has an anatomically shielded microbiome and can modulate and repopulate the microbiome in the rest of the large intestine [14]. Consequently, changes in the appendix microbiome may have a widespread effect on the microbiota of the intestine, which may be reflected in changes in stool microbiota.”

- Line 54: gut microbiota (not exclusively gut microbiota in appendix) have an intimate cross-talk with the immune system in the gut.

Response: We have revised this sentence to more clearly acknowledge that microbiota affect the immune system throughout the gut, while drawing attention to the proximity and density of immune cells within appendix tissue, which has particular relevance for our study.

Page 2 Line 60-64: “Microbiota can modify inflammatory responses and immunity in the gut, and the lymphoid tissue of the appendix is particularly relevant since it is especially rich in lymphocytes compared to the rest of the GI tract [11]; microbial metabolites have direct access to the immune cells within the lymphoid follicles of the appendix [23-25].”

- The hypothesis is not clearly defined. Neither the objective. Authors should not report results or conclusions at the end of the introduction; only hypothesis and objective.

Response: We revised the final paragraph in the introduction to remove the reported results and conclusions and more clearly state our hypotheses and objectives. The text was revised as follows:

Page 2 Line78-82: “To test our hypothesis that the appendix microbiota is altered in PD, with impact on the associated transcription and protein homeostasis in the gut. We performed a multi-omic analysis of the appendix microbiome and appendix tissue, including metabolomics, proteomics, and transcriptomics, in PD patients and controls.

The following text was added to the discussion:

Page 8 lines 201-204: “The microbial taxa and pathway alterations we identified led us to further hypothesize that bile acid metabolism might be directly associated with the etiopathogenesis of PD. Thus, we investigate potential changes in bile acid metabolism in PD using metabolomics, proteomics, and transcriptomics. “

Results

- Line 87: please, avoid citations in this section. Any explanation of the results should be included in the discussion section.

Response: We thank the reviewer for the opportunity to clarify this aspect of our results. We performed a statistical test comparing our results to data from the study cited, and thus left that part in the results section. We revised the sentence to remove the explanation part, and now instead refer to this in the discussion, as suggested by the reviewer.

Page 2 lines 93-96: “We also compared the appendix microbial community identified to that found in surgically-isolated, healthy appendix tissues in a previous study [12] and found significant correlation (order level: R=0.91, p<10-14; family level: R=0.26, p<0.05; Pearson’s correlation).”

Discussion section reads as follows:

Page 8 lines 210-212: “The appendix microbial community identified was similar to that found in surgically-isolated, healthy appendix tissues in a previous study [12], demonstrating reliability in our methods.”

Discussion

- Limitations and strengths of the study are needed.

Response: We revised the second paragraph of our study to more clearly state the strengths and limitations of the study as follows:

Page 8 lines 209-224: “Unlike the prior 16S rRNA and metagenomic studies of the PD microbiome, our metatranscriptomic study has the advantage of capturing only active species. Another strength of our study is the unique location of the survey; microbiota shifts in the appendix are particularly relevant given the role of the appendix in affecting the microbiota in other intestinal regions [11]. The appendix microbial community identified was similar to that found in surgically-isolated, healthy appendix tissues in a previous study [12], demonstrating reliability in our methods. Differences between the appendix and fecal microbiomes limit comparison with previous studies of the PD microbiome; however, an increase in Clostridiales cluster XI has been observed in PD stool previously [36]. Nonetheless, one limitation of our study is that it does not accurately distinguish microbiota changes which may be responsible for causing PD from those that are consequences of the disease. Both constipation (a common symptom of PD) and PD medication may influence the gut microbiota [36]. Though we cannot exclude constipation as a contributing factor, it is unlikely to explain all the changes observed, since increases in both LCA and DCA increase colonic peristalsis and decrease fecal transit time [37,38]. Furthermore, mice given fecal microbiome transplants from constipated donors show a reduction in LCA and DCA, unlike the increase we observed in PD [39]. Although we controlled for age, sex, and postmortem interval, additional clinical variables such as medication or duration of disease were not available for this dataset but would be valuable to test for correlations; we aim to consider this in future work.

Methods

- A first paragraph describing the study design is needed. Details about selection criteria is also needed. Also, preservation methods of tissues in the extraction place and transport should be stated.

Response: We revised the first paragraph to include more details on the study design including tissue extraction and transport, and selection criteria for cases and controls.

Page 9 lines 283-287: “Tissue was collected, and flash frozen within an average postmortem interval of 14 hours and shipped frozen on dry ice. PD cases were selected based on pathologically confirmed presence of brain Lewy body pathology and loss of midbrain neurons, and control individuals were selected for the absence of such pathology. Cases and controls were balanced by sex and age across groups.

- Was the sample size calculated beforehand? If now, which was the rationale for using that and no other sample size?

Response: Sample size was determined based on tissue availability and previously published studies using a similar sample size [1,2]. For metatranscriptomic analysis, we included a minimum of 10 samples per group (PD and control).  For proteomics we used a minimum of 3 per group, and for transcriptomics a minimum of 6 per group; we used a smaller sample size since this part was a confirmation study rather than an initial unbiased analysis. We also balanced age and sex across groups.

[1] Jackson, H.T.; Mongodin, E.F.; Davenport, K.P.; Fraser, C.M.; Sandler, A.D.; Zeichner, S.L. Culture-independent evaluation of the appendix and rectum microbiomes in children with and without appendicitis. PLoS One 2014, 9, e95414, doi:10.1371/journal.pone.0095414.

[2] Li, P., Ensink, E., Lang, S. et al. Hemispheric asymmetry in the human brain and in Parkinson’s disease is linked to divergent epigenetic patterns in neurons. Genome Biol 21, 61 (2020). https://doi.org/10.1186/s13059-020-01960-1

Reviewer 2 Report

The authors report a significant increase in secondary bile acids (LCA and DCA) in the appendix and ileum of Parkinson's disease (PD) patients. This is an interesting manuscript and highlights some potentially important changes relevant to the pathoetiology of PD.

Although the authors seem reluctant to speculate as to the physiological processes that may underpin their data, their may be some value in indicating likely processes to stimulate future research. For example:

There is great interest in the appendix in PD, given that the appendix has a relatively high density of alpha-synuclein [Killinger and Labrie, 2019], suggesting that alterations in the appendix may contribute to the levels of retrograde alpha-synuclein transport to the brain. Data indicates that serum DCA levels are negatively correlated with melatonin [Yue et al., 2019]. It is proposed that a decrease in gut melatonin synthesis contributes to PD pathoetiology, driven by alterations in the microbiome [Anderson et al., 2016]. Melatonin is highly produced in the appendix, as well as the gut, with melatonin suppressing the levels and toxic conformation of alpha-synuclein [Anderson et al., 2016]. Would decreased local melatonin levels in the appendix and gut contribute to levels of inflammatory secondary bile acids, concurrent to increasing alpha-synuclein?

Killinger B, Labrie V. The Appendix in Parkinson's Disease: From Vestigial Remnant to Vital Organ? J Parkinsons Dis. 2019;9(s2):S345-S358. doi: 10.3233/JPD-191703. PMID: 31609697; PMCID: PMC6839473.

Yue S, Zhao D, Peng C, Tan C, Wang Q, Gong J. Effects of theabrownin on serum metabolites and gut microbiome in rats with a high-sugar diet. Food Funct. 2019 Nov 1;10(11):7063-7080. doi: 10.1039/c9fo01334b. Epub 2019 Oct 17. PMID: 31621728.

Author Response

Review 2

The authors report a significant increase in secondary bile acids (LCA and DCA) in the appendix and ileum of Parkinson's disease (PD) patients. This is an interesting manuscript and highlights some potentially important changes relevant to the pathoetiology of PD.

Although the authors seem reluctant to speculate as to the physiological processes that may underpin their data, there may be some value in indicating likely processes to stimulate future research. For example:

There is great interest in the appendix in PD, given that the appendix has a relatively high density of alpha-synuclein [Killinger and Labrie, 2019], suggesting that alterations in the appendix may contribute to the levels of retrograde alpha-synuclein transport to the brain. Data indicates that serum DCA levels are negatively correlated with melatonin [Yue et al., 2019]. It is proposed that a decrease in gut melatonin synthesis contributes to PD pathoetiology, driven by alterations in the microbiome [Anderson et al., 2016]. Melatonin is highly produced in the appendix, as well as the gut, with melatonin suppressing the levels and toxic conformation of alpha-synuclein [Anderson et al., 2016]. Would decreased local melatonin levels in the appendix and gut contribute to levels of inflammatory secondary bile acids, concurrent to increasing alpha-synuclein?

Killinger B, Labrie V. The Appendix in Parkinson's Disease: From Vestigial Remnant to Vital Organ? J Parkinsons Dis. 2019;9(s2):S345-S358. doi: 10.3233/JPD-191703. PMID: 31609697; PMCID: PMC6839473.

Yue S, Zhao D, Peng C, Tan C, Wang Q, Gong J. Effects of theabrownin on serum metabolites and gut microbiome in rats with a high-sugar diet. Food Funct. 2019 Nov 1;10(11):7063-7080. doi: 10.1039/c9fo01334b. Epub 2019 Oct 17. PMID: 31621728.

Response: We thank the reviewer for the positive response to our study and the added insight into the possible physiological processes underlying our results. While decreased melatonin poses a promising pathway by which microbiota changes could drive PD, this does not seem to directly connect to our results; Yue et. al state that serum levels of deoxycholic acid and melatonin were negatively correlated with specific species of gut microbiota (Bacteroides acidifaciens, and Staphylococcus saprophyticus subsp. Saprophyticus), not that DCA and melatonin are negatively correlated to each other; based on this paper it would seem that if the same microbial changes in PD are driving changes in both bile acids and melatonin, then a decrease in melatonin would be accompanied by decreased DCA, not an increase as we saw. Nonetheless, we have added a few sentences to our discussion discussing the potential role of melatonin. We agree that alterations in the appendix are significant given the abundance of pathological alpha-synuclein in the wall of the appendix, and we have revised our results to emphasize this. Another likely physiological process that we have already suggested is that bile acids could contribute to T cell activation, which could be a physiological process underpinning our data that merits further discussion. We have revised the text as follows:

Page 8 lines 239-246: “These effects could propel the accumulation of pathological α-syn aggregates, which could potentially propagate from the gut to the brain through retrograde transport [47,48]. Changes in the appendix are especially relevant since the appendix holds an abundance of α-syn even in healthy individuals [11]. Alterations in the microbiome have been correlated with changes in melatonin synthesis along with serum DCA levels [49]; melatonin, which is highly produced in the appendix, can suppress the accumulation of toxic α-syn and may contribute to PD etiopathogenesis. Thus, local melatonin levels in the PD appendix may impact both the levels of secondary bile acids and α-syn aggregation [50].”

We also expanded the interpretation of our data regarding the physiological processes underlying the increased bile acids as follows:

Page 8 lines 252-262: “In PD patients, there was no evidence of an overproduction of primary bile acids: primary bile acids and total bile acid pool size in the PD ileum remained similar to controls, and we did not detect transcript abnormalities in enzymes responsible for primary bile acid synthesis in the PD liver. However, PD patients may have impaired bile acid reuptake in the ileum, as indicated by the prominent decrease in FABP6, a protein responsible for efficient transport of primary bile acids through enterocytes for recirculation [56]. Thus, disrupted bile acid transport in the ileum and increased bile-metabolizing bacteria in the large bowel combined could be responsible for the elevated levels of secondary bile acids as observed in PD. These disruptions may also contribute to the observed abnormalities in cholesterol homeostasis in the PD gut, as demonstrated by the transcriptional increase in cholesterol transporters and proteomic disruption of lipid metabolism pathways.”

Reviewer 3 Report

This is an exceptionally interesting, well designed study which provides further evidence of a link between microbiome alterations and changes in bile acid profile in Parkinson's disease (PD). The paper is well written, the authors have combined a number of sometimes challenging techniques to support their hypothesis. The results may have considerable implications if confirmed in larger studies. 

However, the discussion should be a little more balanced. In particular: It's perfectly possible that the observed changes are secondary - constipation is of course very common in PD and PD medication is also well known for its effect on the microbiome composition. The authors need to discuss this appropriately.

More clinical information on the included patients (duration of illness, treatment etc) would also be helpful. 

Author Response

Review 3

This is an exceptionally interesting, well designed study which provides further evidence of a link between microbiome alterations and changes in bile acid profile in Parkinson's disease (PD). The paper is well written, the authors have combined a number of sometimes challenging techniques to support their hypothesis. The results may have considerable implications if confirmed in larger studies. 

However, the discussion should be a little more balanced. In particular: It's perfectly possible that the observed changes are secondary - constipation is of course very common in PD and PD medication is also well known for its effect on the microbiome composition. The authors need to discuss this appropriately.

Response: We thank the reviewer for the opportunity to more clearly address the limitations of our study. We note that our study does not distinguish microbiota changes which may be responsible for causing PD from those that are consequences of the disease. We now added an acknowledgement that PD medication may affect microbiome composition. We had also addressed the potential confounder of constipation in a later paragraph, but we moved this sentence to flow more logically earlier in the discussion as we now better discuss the limitations of our study. Constipation may be a contributing factor, but we suggest that it is unlikely to explain all the bile-acid related changes given that increases in both LCA and DCA increase colonic peristalsis and decrease fecal transit time [37,38].

We have revised the paragraph on the limitations of the study as follows:

Page 8 lines 209-224: “Unlike the prior 16S rRNA and metagenomic studies of the PD microbiome, our metatranscriptomic study has the advantage of capturing only active species. Another strength of our study is the unique location of the survey; microbiota shifts in the appendix are particularly relevant given the role of the appendix in affecting the microbiota in other intestinal regions [11]. Differences between the appendix and fecal microbiomes limit comparison with previous studies of the PD microbiome; however, an increase in Clostridiales cluster XI has been observed in PD stool previously [36]. Nonetheless, one limitation of our study is that it does not accurately distinguish microbiota changes which may be contribute to the causes of PD from those that are consequences of the disease. Both constipation (a common symptom of PD) and PD medication may influence the gut microbiota [36]. Though we cannot exclude constipation as a contributing factor, it is unlikely to explain all the changes observed, since increases in both LCA and DCA increase colonic peristalsis and decrease fecal transit time [37,38]. Although we controlled for age, sex, and postmortem interval, additional clinical variables such as medication or duration of disease were not available for this dataset but would be valuable to test for correlations; we aim to consider this in future work.”

More clinical information on the included patients (duration of illness, treatment etc) would also be helpful. 

Response: We agree that additional clinical information would be useful to further interpret our results; however, unfortunately this data was not available from our current tissue source. We will make efforts to obtain such data for future studies and have included a section regarding this matter in our discussion of the studies limitations:

Page 8 lines 222-224: “Although we controlled for age, sex, and postmortem interval, additional clinical variables such as medication or duration of disease were not available for this dataset but would be valuable to test for correlations; we aim to consider this in future work.”